# Kinetics and Mechanisms of Metal Chlorides Catalysis for Coal Char Gasification with CO2

**Yong He, Ye Yuan, Zhihua Wang \*, Longlong Liu, Jiaxin Tan, Jiahao Chen and Kefa Cen**

State Key Laboratory of Clean Energy Utilization, Zhejiang University, Hangzhou 310027, China;
heyong@zju.edu.cn (Y.H.); yuanye0405@zju.edu.cn (Y.Y.); liulonglong@zju.edu.cn (L.L.); tanjiaxin@zju.edu.cn (J.T.); chenjiahao@zju.edu.cn (J.C.); kfcen@zju.edu.cn (K.C.)

\* Correspondence: wangzh@zju.edu.cn; Tel.: +86-571-8795-3162; Fax: +86-571-8795-1616

**Abstract:** The gasification experiments of coal chars with $CO_2$ were carried out isothermally, with K, Ca, Ni, and Zn chloride catalysts, adopting a thermal gravimetric analyzer (TGA) from 800 to 1100 °C. The kinetic characteristic of the samples were described using the volumetric model (VM), the grain model (GM), and the random pore model (PRM). The morphology patterns of the samples were tested applying X-ray diffraction (XRD) and the catalytic mechanisms concerning the phase changes were proposed. The results confirm that the gasification rate and char reactivities are enhanced by K, Ca and Ni chlorides, while $ZnCl_2$ inhibited the process. The catalysis ability shows the following cation order: Ca > K > Ni > Zn. Among the models described above, PRM was proven to give the best fitting value and hence adopted to kinetics parameters calculation. The activation energies in promoting conditions were lower than that of the uncatalyzed cases. In view of the catalytic mechanism, the K metals tend to form intermediate complexes and repeatedly connect with coal char, while the Ca species may follow the oxidation-reduction mechanism and the Ni metals catalyze the gasification process.

**Keywords:** char gasification; catalytic kinetics; catalytic mechanisms; metal chlorides

---

## 1. Introduction

It is generally known that char gasification is the rate-controlling step during coal gasification process [1]. Char gasification with $CO_2$, also called Boudouard reaction, as shown in Equation (1), is a highly endothermic reaction and only achieve high carbon conversion in high-temperature (>700 °C is typically cited) [2]. From both a technological and economical point of view, higher char reactivity and lower reaction temperature are desired. The demands can be reduced by employing catalysts to accelerate the gasification reactions [3,4], which is a prospective mean to achieve a suitable reaction rate and char reactivities, thus mitigating the severe conditions of the gasifiers and reducing the high costs of the process [5].

$$C(s) + CO_2(g) \leftrightarrows 2CO(g) \qquad \Delta H = +172.67 \text{ kJ/mol} \qquad (1)$$

A quantity of research has been conducted using alkali and alkali-earth metals (AAEMs) and transition metals salts as catalysts for the gasification reactions of various chars. Huang at al. [6] studied the catalytic effects of several common metals and discovered that the gasification reactivities of char were improved in the decreasing order: K > Na > Ca > Fe > Mg. A comparison of the effectiveness of three carbonates for $CO_2$ gasification was investigated by Rao at al. [7], and study proved that the catalytic performance of $K_2CO_3$ was better than that of $Na_2CO_3$, $Li_2CO_3$. Lahijani at al. [8] noted that the addition of potassium nitrate salt could deteriorate the sintering tendency, and the char reactivity only got slightly improved. Although these catalysts showed good performance,

expensive prices and the harmful gas decomposed from acid radicals as well as sintering tendency of catalysts make it a necessity to find a cheap and efficient catalyst.

In recent years, metal chlorides have attracted much attention for the catalytic gasification in view of their relatively high reactivities and low prices. Zhou at al. [9] discovered that $FeCl_3$ obtained the best catalytic activity among iron species catalysts in petroleum coke gasification. The catalysis of $FeCl_3$ can be promoted by adding $Ca(OH)_2$ to achieve chloride-free. Very similar results were reported by Takarada at al. [10] who used $NH_3$ and $Ca(OH)_2$ to prepare chloride-free Na and K catalysts, improving the rate 20–30 times of that for raw coal. The rate increase is the same as that by the impregnated carbonate catalysts. Moreover, Encinar at al. [4] confirmed that the presence of LiCl, NaCl and KCl increased the reaction rate, the yield and production of gases in the eucalyptus char steam gasification process. Most previous studies focused on the promoting effects of AAEMs catalysts, therefore, it is essential to provide a comprehensive understanding of the coal char gasification with various kinds of metal chloride catalysts. Moreover, there is still a lack in terms of catalytic mechanisms of metal chlorides, which needs further exploration.

In this study, the catalysis performances of four metal chlorides, including alkali metal (K), alkali earth metal (Ca), and transitions metals (Zn and Ni), on lignite char gasification were examined by TGA. On the basis of TG profiles, the three most common reaction models were adopted to understand the reaction kinetics. Moreover, XRD was conducted to clarify the catalytic mechanisms in the coal char gasification.

## 2. Results and Discussion

### 2.1. Influence of Reaction Temperature on Char Gasification

Figure 1 presents the influence of temperature on RAW coal and AW （acid-washing） coal char gasification. Two samples bear a resemblance that increasing temperature drastically shortens the reaction time and this result is foreseen because, in fact, Boudouard reaction is favored at high temperatures. In detail, it is clear that the reaction proceeds are very slow at low temperature (800 °C) and complete conversion is reached even more than 900 min and 260 min for AW and RAW samples. While with a 100 °C increase in temperature, completion of char reaction is achieved in approximately 100 min and 50 min respectively. These results demonstrate that the reaction temperature plays a dominant role in gasification kinetics.

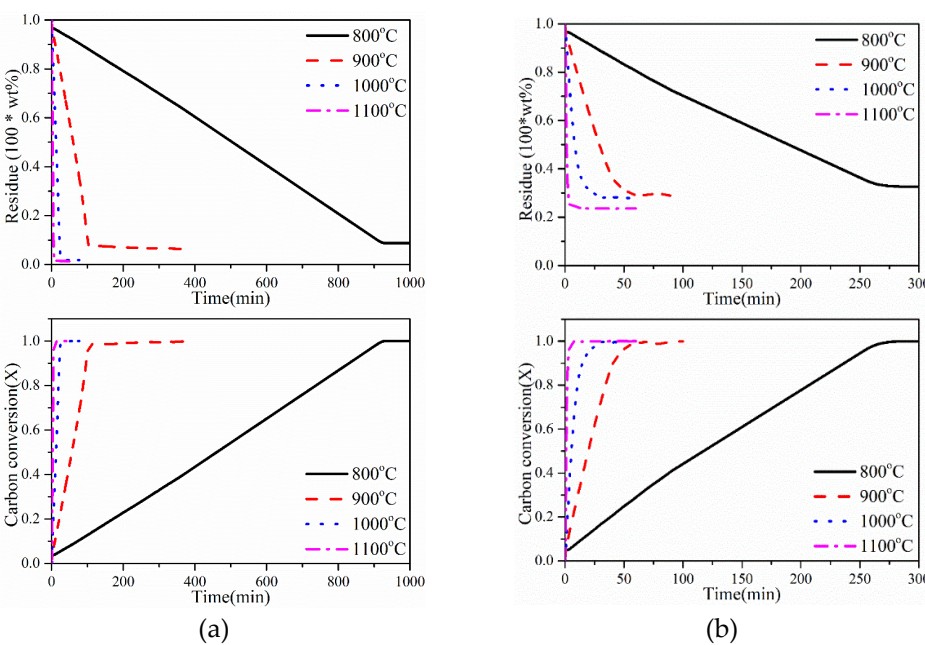

**Figure 1.** Effects of temperatures on gasification performance of RAW and AW coals with thermo gravimetric and carbon conversion. (**a**) AW, (**b**) RAW.

Table 1 shows the change of gasification characteristic parameters as a function of reacting temperature. It can be found that these parameters are consistent with the changing trend of the curve in Figure 1. With increasing temperature, the $\tau_{50}$ time is getting shorter and a higher reactivity index is achieved, meaning a faster gasification rate. Meanwhile, the reactivity index of AW char increases by a large margin than RAW char. The reactivity index of AW char in 1000 °C is 37 times of that in 800 °C, but it is only 20 times for RAW char, which implies that the temperature has a deeper effect during the gasification process in terms of change in gasification rate rather than other factors.

**Table 1.** Effects of temperature on gasification characteristic parameters.

| Sample | $T$/$^{\circ}$C | $\tau_{50}$/min | R |
|---|---|---|---|
| | 800 | 117.50 | 0.0043 |
| RAW | 900 | 19.80 | 0.0253 |
| | 1000 | 5.60 | 0.0893 |
| | 1100 | 1.15 | 0.4348 |
| | 800 | 460.30 | 0.0011 |
| AW | 900 | 56.20 | 0.0089 |
| | 1000 | 12.20 | 0.0410 |
| | 1100 | 2.85 | 0.1754 |

It is worthy of notice that the difference of conversion degree from 1000 to 1100 °C are negligible compared to lower temperatures, which can be explained by the change of reaction control regime. It is considered that the process is under control of chemical reaction when the temperature is under 1000 °C, whereas the synergy of chemical reaction control and pore diffusion regime plays a part in 1100 °C. Some researchers [11–13] have reached the same conclusion that 1000 °C is the critical temperature in the $CO_2$ gasification of coal char.

## 2.2. Influence of Various Catalysts on Char Gasification

To better evaluate the catalytic efficacy of different metal chlorides in gasification, the TGA data versus time measured at the same final temperature are shown in Figure 2. Results show that the catalysis of metal chlorides exhibits different behaviors. As gasified at 800 °C, the time needed to achieve complete conversion for AW coal char was approximately 900 min, however it is only 50 min and 130 min for AW-$CaCl_2$ and AW-KCl char, which implies that the AW gasification rate is 18 and 7 times slower than catalytic gasification rate respectively. The presence of $CaCl_2$ and KCl greatly enhances the gasification rate and sharply shortens reaction time. Extensive research [4,14,15] has confirmed that AAEM as catalysts show outstanding performance in both pyrolysis and gasification. Contrarily, the additive of $ZnCl_2$ inhibits the catalytic gasification when the temperature is above 900 °C, contributing to lower gasification reactivity than AW sample. As for the use of $NiCl_2$, the weight loss curve is far akin to RAW coal char sample, which indicates that the $NiCl_2$ catalysts act as the same promoting function with inherent minerals in Raw sample. All results obtained above suggest that the gasification reactivity follows the anions order as: Ca > K > Ni > Zn. Such a trend is quite similar to those represented in the research for the catalytic gasification of char samples with $CO_2$ [6,16]. So the catalytic ability of metal chlorides can be divided into two groups, one group including $CaCl_2$, KCl, and $NiCl_2$ positively catalyzes the reaction, while another such as $ZnCl_2$ hinders the gasification process.

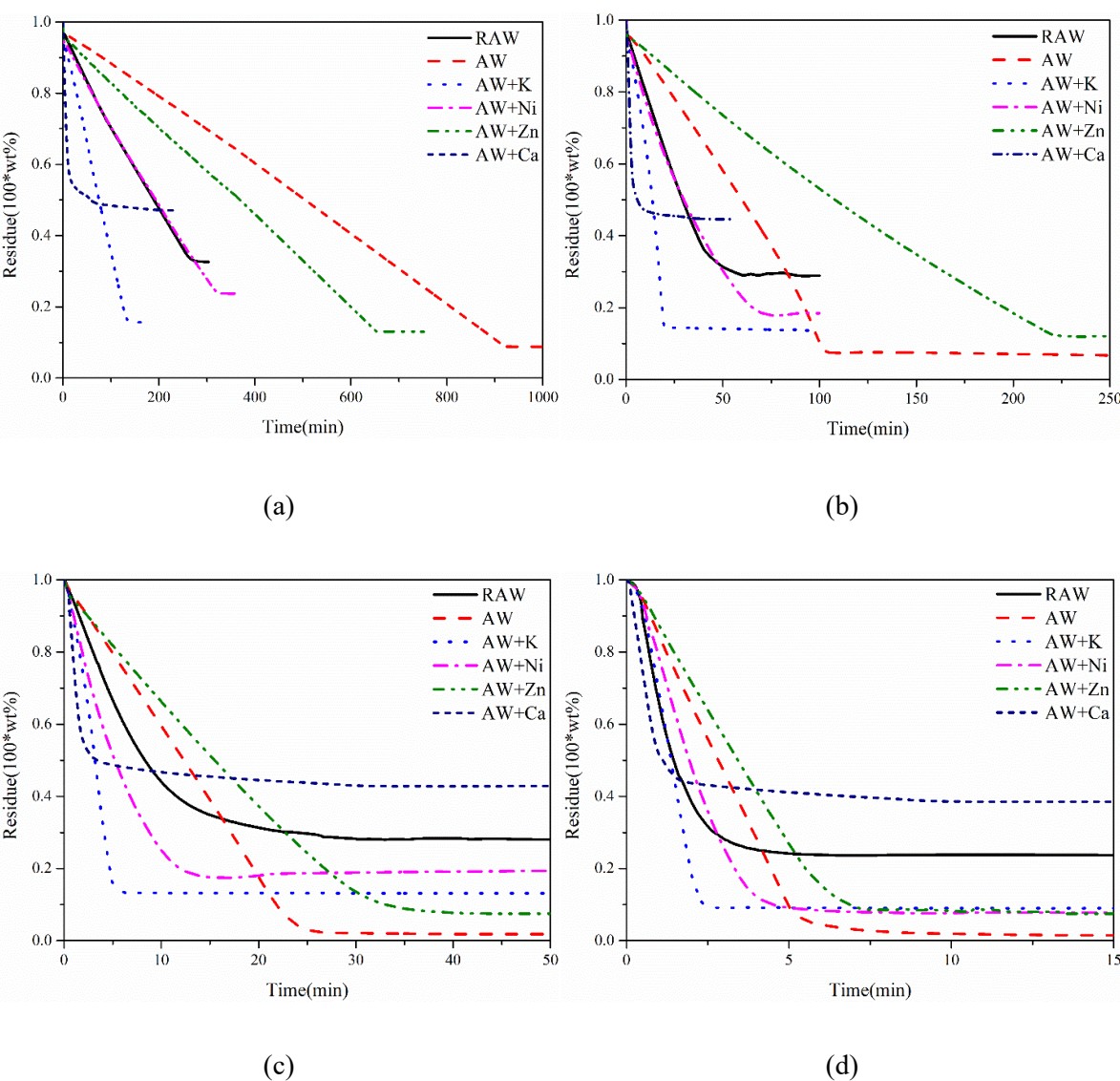

**Figure 2.** Effects of different metal chlorides on AW char gasification. (**a**) 800 °C;(**b**) 900 °C; (**c**) 1000 °C; (**d**) 1100 °C.

Concerning the mass of remaining residue after the gasification process, there are evident differences between AW samples and others. It is anticipated that the catalysts remain in different forms after reaction, while the inherent minerals only account for 0.52% in AW sample after acid-washing, thus almost complete conversion of AW coal char can be observed. It is of interest the residual amount of AW-CaCl$_2$ continues to be very high compared with other four samples. Matsukata at al. [17] suggest that catalysts, such as calcium, were hard to vaporize and diffuse into the bulk, most of which kept constant on the char surface. There was no appreciable variation in Ca content at any carbon conversion level. In this case, CaCl$_2$ catalysts in the residue are left after the reaction. We note that the residuary amount of three samples with K, Zn, Ni catalysts after the reaction is fairly close, reveling that the metal chlorides can facilitate the gasification rate, but the final conversion is dependent on the intrinsic characteristics of char rather than these catalysts.

Moreover, to further elucidate the mechanism of catalysis, Figure 3 demonstrates the reactivity data via carbon conversion at 1000 °C. It is obvious that the Ca and K exist in the coal chars largely enhance the char reactivity and change the reactivity profiles; this character is not noticeable with AW-Ni, -Zn chars. Du at al. [18] points out that the catalysis species rather than active structures control the char reactivity with conversion, thus it is considered that the addition of metal chlorides alters reactivity profiles.

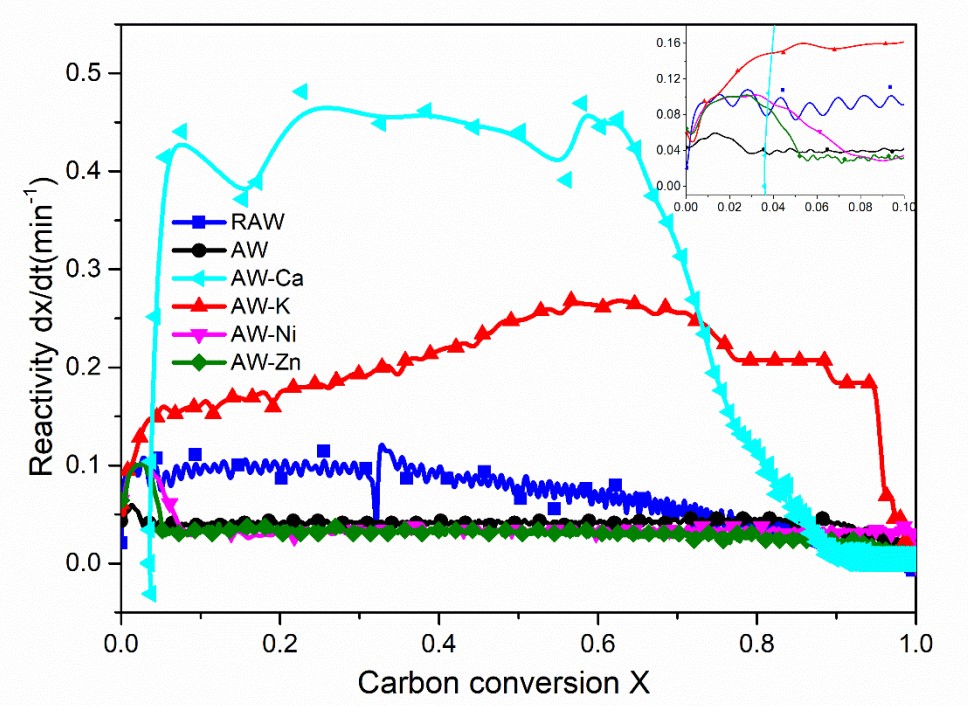

**Figure 3.** Gasification reactivity of different added metal chlorides versus conversion profiles at 1000 °C (The insert figure is a zoom of the carbon conversion at 0–0.1).

The reactivity of KCl-char is lower than that of CaCl$_2$-char at conversion ranges of 0.04 to 0.7, when the carbon conversion reaches a eigenvalue ($X_i$ = 0.7), the reactivity of KCl-char becomes the higher one. The shift in the X value suggests that CaCl$_2$ and KCl show different mechanisms during CO$_2$ gasification. Previous work has proved that the dispersion conditions [19,20] and the mobility of the catalysts [21,22] are significant properties for catalytic reactivities. Therefore, the good dispersion and the persistence of Ca species at high temperature both contribute to a sharp increase in char reactivity. While the saturation of Ca species in high conversion will promote sintering of Ca resulting in a poor dispersion of catalysts, corresponding to the remarkable decrease in reactivity, inducing lower reactivity than KCl-char. In contrast, the loss of KCl catalysts is the dominant factor for the deactivation process and the K metals are prone to vaporize and easily migrate into the bulk of carbon, increasing the reactivity and then reaching a maximum with the higher conversion. Consequently, the reactivity of KCl-char maintains at a relatively high level until the end of gasification.

## 2.3. Kinetic Analysis

### 2.3.1. Reaction Rate Calculation

In our work, a carbon conversion range from 0.1 to 0.7 was employed to analyze the reaction rate, because within a high conversion range the breakdown of the porous structure brings about the domination of the ash layer to the completion of the reaction process. Figure 4 presents the fitting effects between three models applied and experimental results obtained for the catalytic gasification at different temperatures. All samples exhibit a similar pattern that the slope of the alignment lines become deeper with higher temperatures. According to Equations (7)–(9), the slope of the fitted line is the apparent reaction rate constant. Therefore, the higher temperature, the greater reactivity, which is consistent with experimental results in Section 2.1.

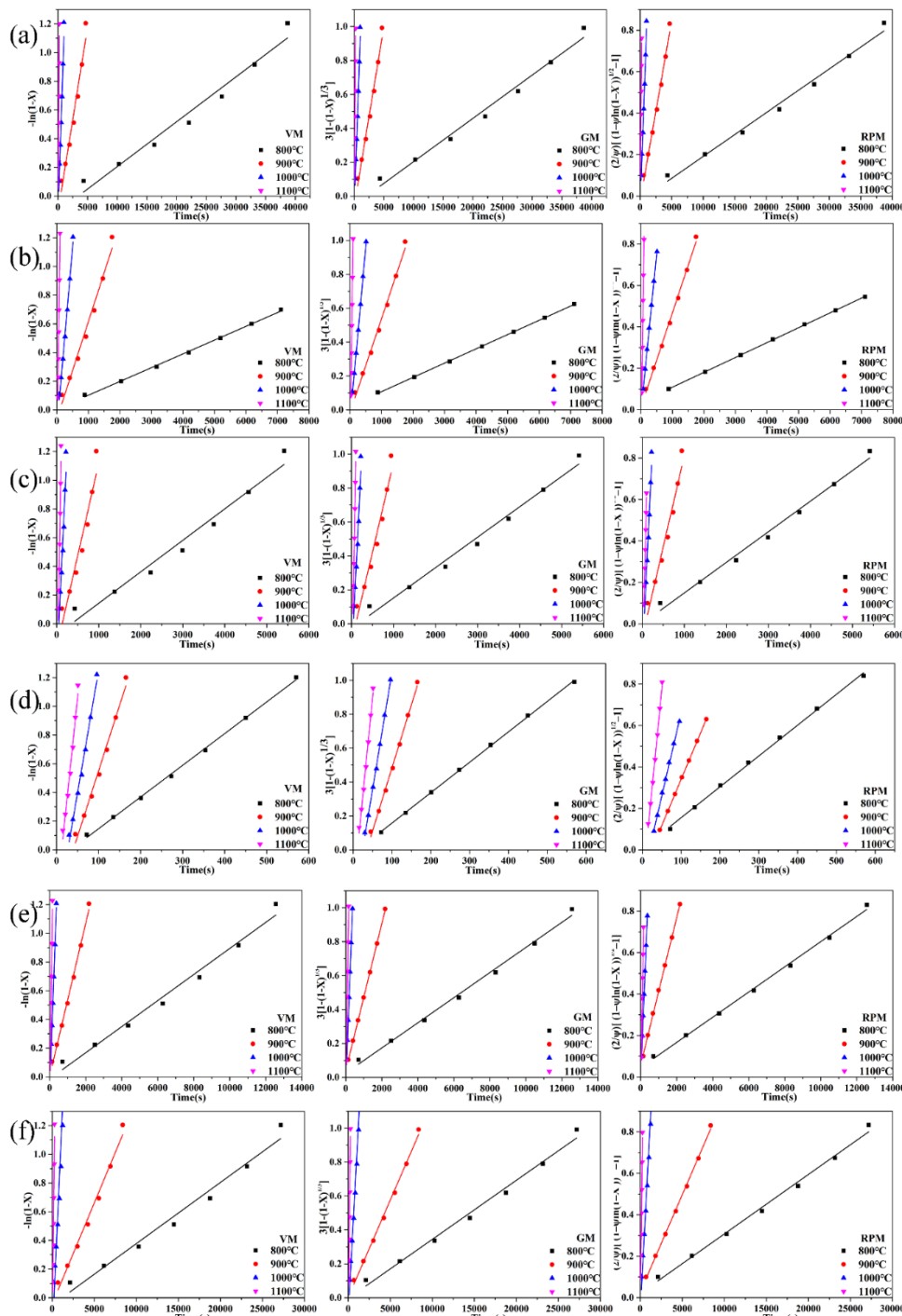

**Figure 4.** Plots fitting of VM, GM and RPM for (**a**) AW, (**b**) RAW, (**c**) AW-KCl, (**d**) AW-CaCl₂, (**e**) AW-NiCl₂, (**f**) AW-ZnCl₂.

For a better understanding and analysis of the results presented in Figure 4, Table 2 summarizes the kinetic parameters and the coefficient of determinations ($R^2$) for three models. Results show that PRM has the highest values of $R^2$, most of which is higher than 0.99, followed by the results of GM. On the contrary, the worst fitting can be observed for VM since this model assumes a simplistic reaction mechanism, causing considerable deviation from the actual reaction process. It is of interest that the reaction rate derived from three models is subject to the following decreasing orders: VM > GM > PRM, for the same sample at any temperature. This can be explained by the theory of reaction

models, the effective area in the assumption gradually decreases, matching the decreasing reaction rate for the models.

**Table 2.** The kinetic and empirical parameters of three employed models.

| Char | T (°C) | VM | | GM | | RPM | | |
|------|--------|-----------|--------|-----------|--------|-----------|--------|--------|
| | | K (s⁻¹) | R² | K (s⁻¹) | R² | K (s⁻¹) | R² | ψ |
| AW | 800 | 0.00003124 | 0.9640 | 0.00002552 | 0.9825 | 0.00002111 | 0.9885 | 2.1117 |
| | 900 | 0.0002589 | 0.9642 | 0.0002114 | 0.9827 | 0.0001739 | 0.9888 | 2.1555 |
| | 1000 | 0.00124 | 0.9679 | 0.00101 | 0.9853 | 0.0008424 | 0.9905 | 2.0497 |
| | 1100 | 0.00573 | 0.9760 | 0.00467 | 0.9904 | 0.00348 | 0.9962 | 3.0219 |
| RAW | 800 | 0.00009593 | 0.9978 | 0.0000841 | 0.9997 | 0.00007164 | 0.9999 | 2.0953 |
| | 900 | 0.0006796 | 0.9783 | 0.0005540 | 0.9920 | 0.0004568 | 0.9959 | 2.1279 |
| | 1000 | 0.00253 | 0.9941 | 0.00205 | 0.9995 | 0.00153 | 0.9996 | 3.0356 |
| | 1100 | 0.0165 | 0.9977 | 0.01337 | 0.9996 | 0.0106 | 0.9974 | 2.4231 |
| AW-K | 800 | 0.0002193 | 0.9647 | 0.000179 | 0.9825 | 0.0001476 | 0.9888 | 2.1406 |
| | 900 | 0.00129 | 0.9199 | 0.00106 | 0.9484 | 0.0008754 | 0.9592 | 2.1109 |
| | 1000 | 0.00618 | 0.9305 | 0.00507 | 0.9567 | 0.00418 | 0.9669 | 2.1479 |
| | 1100 | 0.01675 | 0.9636 | 0.01361 | 0.9823 | 0.008 | 0.9950 | 6.1389 |
| AW-Ca | 800 | 0.00221 | 0.9983 | 0.00179 | 0.9999 | 0.00149 | 0.9982 | 2.0561 |
| | 900 | 0.00913 | 0.9852 | 0.00743 | 0.9957 | 0.00447 | 0.9999 | 5.7423 |
| | 1000 | 0.01703 | 0.9913 | 0.01382 | 0.9984 | 0.00807 | 0.9982 | 6.2460 |
| | 1100 | 0.02809 | 0.9825 | 0.02294 | 0.9934 | 0.01904 | 0.9965 | 2.0615 |
| AW-Ni | 800 | 0.00009086 | 0.9809 | 0.00007404 | 0.9936 | 0.00006074 | 0.9969 | 2.1670 |
| | 900 | 0.0005335 | 0.9951 | 0.0004331 | 0.9998 | 0.0003558 | 0.9995 | 2.1446 |
| | 1000 | 0.00342 | 0.9966 | 0.00277 | 0.9999 | 0.0021 | 0.9986 | 2.8509 |
| | 1100 | 0.0096 | 0.9858 | 0.0078 | 0.9963 | 0.00538 | 0.9999 | 3.8698 |
| AW-Zn | 800 | 0.00004264 | 0.9722 | 0.00003479 | 0.9882 | 0.00002867 | 0.9929 | 2.1365 |
| | 900 | 0.0001407 | 0.9842 | 0.0001145 | 0.9955 | 0.00009396 | 0.9982 | 2.1659 |
| | 1000 | 0.0009722 | 0.9276 | 0.0009324 | 0.9754 | 0.0007684 | 0.9767 | 2.0800 |
| | 1100 | 0.00494 | 0.9750 | 0.00402 | 0.9900 | 0.00314 | 0.9954 | 2.5999 |

The calculated reaction rate constant with different catalysts using kinetic models provides a quantitative insight into the comparison of catalysis. As shown in table 2, except $ZnCl_2$-char, char samples with metal chlorides have a higher reaction rate constant than AW-char, indicating the promoting effect of these catalysts. Taking the reaction rate constant in 800 °C and employing the VM model as an example, the value of AW-char is $3.124 \times 10^5$ s⁻¹, after the addition of $CaCl_2$, KCl and $NiCl_2$, it is 700, 7, and 3 times of AW-char respectively. These quantitative data agrees with the conclusion made by qualitative analysis in the last section.

### 2.3.2. Kinetic Parameters Calculation

For the determination of the kinetic parameters, the Arrhenius law presented in Equation (16) was used with PRM model at different temperatures. Then the activation energy (*E*) and the pre-exponential factor (*k*) were calculated from the Arrhenius relationships, but this method only works for the data achieved under control by chemical reaction regime. It can be detected from the changes in the line slope between the chemical and diffusional controlled regime. As reviewed in Section 2.1, the reaction above 1000 °C is within the transition zone, where the chemical reaction at the char surface and mass transfer in the pores jointly control the gasification rate. An acceptable liner relationship is observed as shown in Figure 5, and all data fit well into the Arrhenius plot in the studied temperature range, implying that the gasification reaction is mainly chemically controlled. Based on the discussion above, the two kinetic parameters were calculated and included in Table 3.

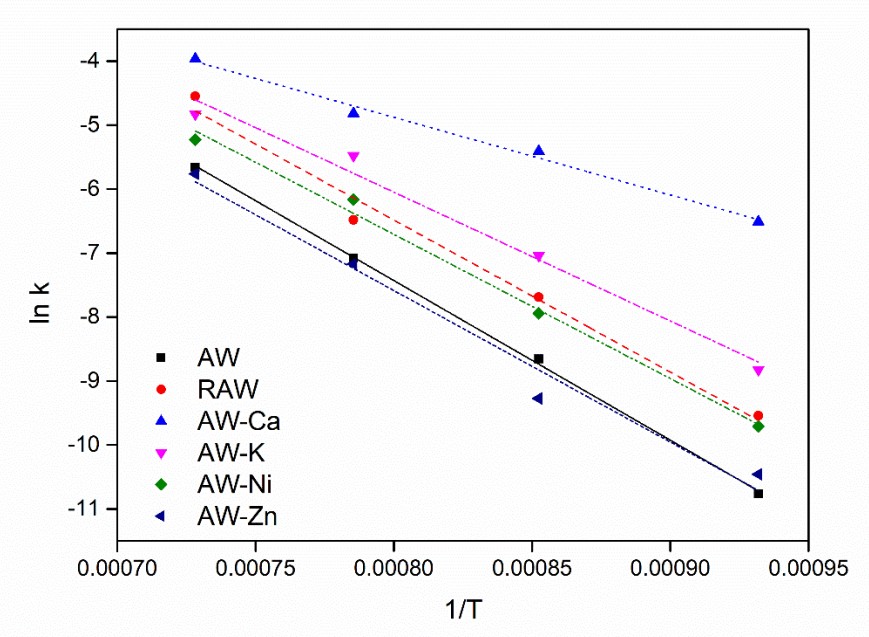

**Figure 5.** Arrhenius plot of char samples using RPM model.

**Table 3.** The kinetic parameters of all char samples.

| Sample | E (kJ/mol) | A (s⁻¹) | R² |
|--------|-----------|---------|-----|
| AW | 207.53 | $2.79 \times 10^5$ | 0.9990 |
| RAW | 197.57 | $2.76 \times 10^5$ | 0.9797 |
| AW-K | 167.46 | $2.35 \times 10^4$ | 0.9777 |
| AW-Ca | 100.99 | $1.26 \times 10^2$ | 0.9879 |
| AW-Ni | 187.41 | $8.29 \times 10^4$ | 0.9914 |
| AW-Zn | 210.03 | $8.69 \times 10^4$ | 0.9683 |

It can be observed in Table 3 that the activation energy varied notably for different samples. To the exclusion of AW-Zn char, the AW-char showed the highest activation energy values, however, this value significantly decreases with the utilization of catalysts. The activation energy follows the decreasing order: AW-Zn > AW > RAW > AW-Ni > AW-K > AW-Ca, which is consistent with the trend described in 2.2. This result confirms that the employment of metal chlorides mitigates harsh conditions and lowers the reaction temperature, making the reaction between coal char and gasifying agent easier.

*2.4. Mechanism of Catalytic Gasification*

The phase of metal chloride has changed due to the preparation of char by pyrolysis at 1000 °C, then the new phase may catalyze gasification process. In the previous study [15,23], the morphs of KCl, CaCl₂, and NiCl₂ are confirmed to become KCl, CaCO₃ and CaCl₂, Ni after pyrolysis respectively, resulting in the different mechanism of each sample. XRD measurements were carried out to find out the mechanism of each catalyst, and the spectra of three samples after gasification at 1000 °C is exhibited in Figure 6. In the gasification reaction, the migration and transformation of K species largely occurred. It can be seen from Figure 6(a), most peaks of KCl disappeared in the sample after gasification, meaning the consumption and deactivation of the catalysts. While the other crystallites of K species were formed during the process, including $C_nK$ and $K_2CO_3$. A reaction cycle based on the Electron Donor-Acceptor (EDA) complexes proposed by Wen at al. [24] could account for this formation. As soon as the K metals are generated in the gasification process, some EDA complexes $C_nK$ (n = 8, 16, 24) will be formed to catalyze the reactions with $CO_2$:

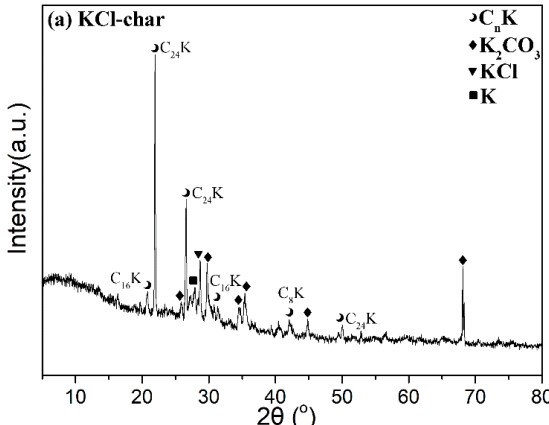

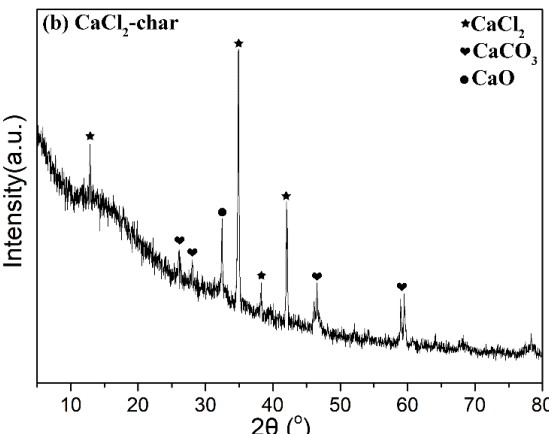

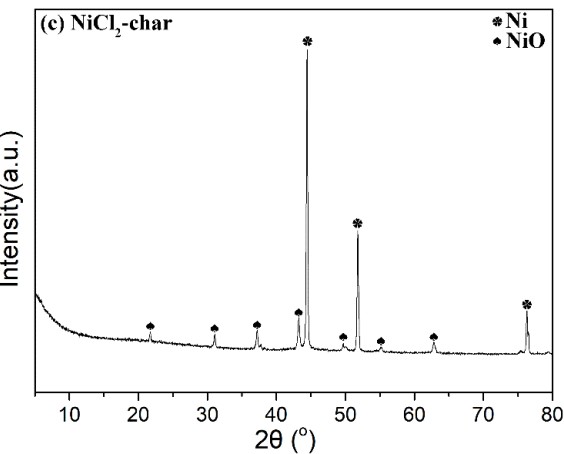

**Figure 6.** XRD patterns of the studied samples after gasification, (**a**) KCl-char, (**b**) CaCl₂-char, (**c**) NiCl₂-char.

$$K_2CO_3 + 2C = 2K + 3CO \tag{2}$$

$$2K + 2nC = 2C_nK \tag{3}$$

$$2C_nK + CO_2 = (2C_nK) \cdot OCO = 2nC \cdot _2O + CO \tag{4}$$

$$(2nC) \cdot K_2O + CO_2 = (2nC) \cdot K_2CO_3 = 2nC + K_2CO_3 \tag{5}$$

$$\text{Overall: } C + CO_2 = 2CO \tag{6}$$

The peaks of $CaCl_2$, $CaCO_3$, and CaO are observed in the Figure 6b, and the $CaCl_2$ first decomposes with water in coal as shown in Equation (7), then the resulting CaO tends to react with $CO_2$ to generate $CaCO_3$.

$$CaCl_2 + H_2O = CaO + 2\,HCl \tag{7}$$

The mechanism of the sample can be explained by oxidation-reduction mechanism [7], which involves the following reactions:

$$CaCO_3 + 2C = Ca + 3CO \tag{8}$$

$$Ca + CO_2 = CaO + CO \tag{9}$$

$$CaO + CO_2 = CaCO_3 \tag{10}$$

$$\text{Overall：} \ C + CO_2 = 2CO \tag{11}$$

It is inferred that the phases of $CaCl_2$ and $CaCO_3$ both act as catalysts in the process. Therefore, the catalytic effects of $CaCl_2$ is better than KCl with single mechanism, but the sintering of carbonates hinders the further catalysis in the later gasification process.

As for $NiCl_2$-char sample, spectra of Ni and NiO are found in the XRD results after the gasification. This may be described by the following reaction:

$$Ni + CO_2 = NiO + CO \tag{12}$$

$$NiO + C = Ni + CO \tag{13}$$

$$\text{Overall: } C + CO_2 = 2CO \tag{14}$$

It is noteworthy that the lack of C atoms is unable to fully support the reduction reaction of NiO, conducing the coexisting of Ni/NiO. So, the Ni atoms and NiO synergistically work as catalysts in the reaction.

## 3. Materials and Methods

### 3.1. Char Preparation

Pingzhuang coal, a typical lignite in Inner Mongolia of China was used in this paper. Raw coal was crushed and sized to meet the particle size requirements (<75 μm). To remove the interferences of inherent minerals in coal ash, the original coal samples were demineralized by acid-washing process (HCl and HF). The details of acid washing procedure can be found in our previous works [23,25]. The coal sample after treatment was named as AW and the raw sample was named as RAW. Table 4 shows the proximate and ultimate analysis of Pingzhuang coal, and the ash analysis result of RAW samples is shown in Table 5. After acid-washing treatment, it is apparent that the ash content decreases to 0.52 wt. %. Therefore, the influence of inherent minerals in AW samples can be ignored in the next gasification study.

**Table 4.** Proximate and ultimate analysis of Pingzhuang coal.

|  | RAW | AW |
|---|---|---|
| Proximate analysis (wt. %) |  |  |
| Ash (db [a]) | 13.00 | 0.52 |
| Volatile (daf [b]) | 48.19 | 45.71 |
| Ultimate analysis (wt.%, daf) |  |  |
| Carbon | 71.32 | 70.29 |
| Hydrogen | 4.13 | 5.37 |

| | | | 1.36 | 1.46 |
|---|---|---|---|---|
| Nitrogen | | | 1.36 | 1.46 |
| Sulfur | | | 0.79 | 0.51 |
| Oxygen [c] | | | 22.40 | 22.37 |

[a] db = oven dry basis, [b] daf = dry and ash free. [c] Calculated by difference.

**Table 5.** Ash compositions (wt. %) of RAW coal.

| $SiO_2$ | $Al_2O_3$ | $Fe_2O_3$ | CaO | MgO | $K_2O$ | $Na_2O$ |
|---|---|---|---|---|---|---|
| 57.3 | 16.32 | 6.15 | 4.57 | 4.93 | 1.24 | 2.68 |

An incipient wetness impregnation method was applied to mix AW samples with metal chlorides. Analytical reagent KCl, $CaCl_2$, $NiCl_2$, $ZnCl_2$ (99.99%, Aladdin, Shanghai, China) were separately added to the samples via co-slurrying with deionized water, according to the ratio of 6 wt. %. All slurry samples were stirred for 24 hours using a magnetic mixer, then drying at 80 °C with 12 hours retention in a vacuum oven, then the samples are designate as AW-K, AW-Ca, AW-Ni, and AW-Zn respectively.

In order to diminish the influence of devolatilization process on the gasification characteristic of coal chars, the samples with volatiles removed were prepared in advance. The samples were subjected to pyrolysis in a box-type protective atmosphere furnace (HMX1600-30, Wuhan, China) under neutral atmosphere of pure nitrogen. The pyrolysis process is as follows: the reacting temperature is raised from room temperature to 1000 °C, with a heating rate of 5 K/min, followed 30-min retention time, then cooling to ambient temperature under nitrogen protection. The whole process was executed at a flow rate of 0.5 L/min under $N_2$ atmosphere.

### 3.2. TGA Experiment

Kinetic experiments were isothermally performed in a TGA (NETZSCH STA 449F3, Selb, German). Experiments were conducted at four temperatures: 800 °C, 900 °C, 1000 °C, and 1100 °C. A weighed char sample (15mg ± 0.1mg), particle sizes smaller than 75 μm, was loaded into a crucible inside the furnace. The sample was heated up to reaction temperature with an increasing rate of 20 °C/min under a nitrogen stream of 70 mL/min. This flow rate was maintained for 90 min to ensure no longer changes in the sample mass. Once the pyrolysis process was finished, the isothermal gasification of the sample was activated by turning on the $CO_2$ (70 mL min$^{-1}$) and maintained at this temperature until complete gasification. A schematic sketch of the gasification reaction process was illustrated in Figure 7.

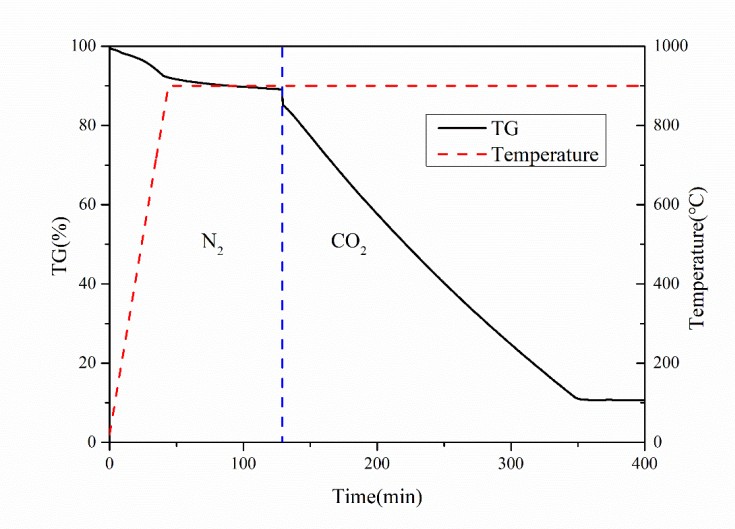

**Figure 7.** Schematic sketch of the reaction process.

## 3.3. Kinetic Models

The overall apparent reaction rate is represented by [26]:

$$r = dX/dt = k(T, Pco_2)f(X) \tag{15}$$

where $k$ is the reaction rate, dependent on reacting temperature $T$ and $CO_2$ partial pressure ($CO_2$ concentration), and $f(X)$ has a structural meaning and corresponds to carbon conversion. Presuming that the $CO_2$ partial pressure keeps constant during the gasification process, the reaction rate (k) can be parameterized according to Arrhenius relationship:

$$k = Aexp(-E/RT) \tag{16}$$

where *A, E,* and R are the pre-exponential factor, the activation energy (kJ/mol), and the universal gas constant 8.314 J/(mol K), respectively.

In this study, three most popular one-step reaction kinetics models were applied to define *f(X)*: the volumetric model (VM), the grain model (GM), and the random pore model (PRM).

VM is the simplest model, assuming a homogenous reaction throughout the entire char particle [27,28] and is represented by:

$$dX/dt = k_{VM}(1-X) \tag{17}$$

GM assumes that porous particles are made up of an assembly of homogeneous non-porous grains and the reaction takes place at the external surface of grains with a spherical shape of the porous [29]. The model is given by:

$$dX/dt = k_{GM}(1-X)^{2/3} \tag{18}$$

PRM assumes that the random overlaps of pore structures could reduce the available areas as reaction progresses [30]. The reaction rate is represented by:

$$dX/dt = k_{PRM}(1-X)\sqrt{1 - \Psi \ln(1-X)} \tag{19}$$

where $\Psi$ is the structural parameter of pore surface and a simple calculation method is to find maximal conversion rate *Xmax* as shown in:

$$\Psi = 2/(2\ln(1-X_{max}) + 1) \tag{20}$$

Table 6 summarizes the linearized solution of VM, GM, and PRM after separating variables and integrating Equations (17)–(19).

**Table 6.** The linearized solution of different models.

| Model | Linearized solution |
|-------|---------------------|
| VM | $k_{VM}t = -\ln(1-X)$ |
| GM | $k_{GM}t = 3[1-(1-X)^{1/3}]$ |
| PRM | $k_{PRM}t = (2/\Psi)[\sqrt{1-\Psi\ln(1-X)} - 1 - 1]$ |

## 3.4. Char Characteristics

The crystalline diffraction peaks of char samples after gasification were checked by X-ray diffraction (XRD, X-pert Powder, Panalytical B.V., Almelo, Holland) to conclude the catalytic mechanisms of metal chloride catalysts. XRD measurement consisted of using a Cu anode at 40 kV and 40 mA and the scanning range was obtained over a $2\theta$ = 5 to 80° with rate of 0.02°/s.

## 4. Conclusion

The catalytic kinetics and mechanisms on metal chlorides of coal char gasification with $CO_2$ have been comprehensively investigated. The main findings can be summarized as follows:

The completion reaction time and $\tau_{50}$ time are drastically shortened with the increasing temperature, which corresponds to a higher reactivity index. The critical temperature for Pingzhuang

coal char gasification is approximately 1000 °C, under which the process is controlled by the chemical reaction regime.

The addition of KCl, CaCl₂, and NiCl₂ enhances the gasification rate and alters the reactivity profiles. In contrast, the presence of ZnCl₂ inhibits the process. Further, the catalytic ability is as follows: Ca > K > Ni > Zn.

All three models give acceptable results for the prediction of reaction rate, and RPM has the best fitting condition with $R^2$ values around 0.99.

K species tend to form Electron Donor-Acceptor complexes and repeatedly connected with the coal/char matrix, while the effect of Ca metal can be explained by the oxidation-reduction mechanism, as for the Ni metal, the coexisting of Ni/NiO synergistically catalyzes the reactions.

**Author Contributions:** Conceptualization, Z.W. and L.L.; methodology, Z.W.; software, L.L. and Y.Y.; validation, Z.W., Y.H. and J.T.; investigation, Y.Y. and J.T.; resources, K.C.; data curation, L.L. and J.C.; writing—original draft preparation, Y.Y.; writing—review and editing, Z.W., Y.H. and J.T.; funding acquisition, Z.W. All authors read, revised and approved the manuscript.

**Funding:** This work was supported by the National Natural Science Foundation of China (51776185 and 51621005) and the Fundamental Research Funds for the Central Universities (2019XZZX005-1-01).

**Conflicts of Interest:** The authors declare no conflict of interest.

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
