# Peer review of "Kinetics and Mechanisms of Metal Chlorides Catalysis for Coal Char Gasification with CO2"

_catalysts, doi:10.3390/catal10060715_

Round 1

Reviewer 1 Report

This manuscript reports an interesting contribution on the effect of homogeneous catalysts on the gasification reaction. This work appears as an incremental work with respect to previous results; however, it reports novel results that can deserve publication after the modifications required.

  • First, check the template for manuscript preparation. The standard sectioning is: introduction, results, discussion (mandatory), materials and methods, conclusions.
  • Samples show different residues after gasification. Did the authors calculate the minimum residue for each sample? Moreover, analysis (elemental, XRD, etc.) of residues would improve the knowledge of the phenomena occurring during gasification.
  • Generally speaking, when chlorides are used, the fate of chlorine is critical, because it is a harmful pollutant. No idea of the chlorine compounds produced during gasification is reported.
  • Figure 5. Except few cases, all the curves at 800 °C show a more than linear behaviour rather that a linear one. Please explain.
  • Page 2, lines 74 and 77. No results on Mn-catalyzed sample are reported. Please check.
  • In figure 3 legends are not fully visible. This prevents the reader from coupling the sample and the experimental curve.
  • Figure 4. The superposition of cyan curve on the inserted frame reduces readability.

Author Response

Dear reviewer,

We response the comments point-by-point and upload a pdf file.

Reviewer 2 Report

The study evaluates kinetics and mechanism of metal chlorides catalysis for coal char gasification with CO2. It demonstrates that K, Ca and Ni chlorides enhance gasification rate and char reactivities, which is a practical conclusion. The manuscript itself is quite well organized, well written and easy to follow. Therefore, the manuscript can be accepted for the publication in Catalysts journal. As namely the technical quality could be further improved, minor revision prior to its publication is recommended. Specific comments and suggestions are given below.

Despite the meaning of the title is clear, I suggest (at least) to replace “on” by “for” (… for gasification of coal char with CO2) or to modify it (e.g.) to: “Kinetics and mechanism of metal chlorides catalysis for coal char gasification with CO2”.

Please check the numbering of equations and chemical reactions (as there are no numbers 2, 3, 4 etc.)

3.3.2. Kinetic parameters calculation. Line 243: “Then the activation energy (Ea) and the pre-exponential factor (ko) were calculated from Arrhenius  ….” The meaning is clear but the symbols are not exactly consistent with those used in Eq. (6) (line 105) where pre-exponential factor A and activation energy E is used. It should be consistent also with other symbols used throughout the manuscript, e.g. in Table 6 (now, Ea is used there). Please check the whole manuscript.

Figure 3 (a-d): Please revise these diagrams because now the legends are not readable (the metal chlorides corresponding to individual curves are not displayed there)

Line 182: “Matsuoka [24]”; line 191: “Du [25]” etc. should be expanded to “Matsuoka et al. [24]” and “Du et al. [25]”. Please check the whole manuscript for similar corrections.

Conclusion is quite well written; perhaps, please use the full expression for “EDA”.

Author Response

(The authors gave the same response as above.)

Round 2

Reviewer 1 Report

The revised manuscript deserves publication.